# Etiology, characteristics and occurrence of heart diseases in rural Lesotho (ECHO-Lesotho): A retrospective echocardiography cohort study

Nzola John Ndongala[1], Callixtina Maepa[1], Emmanuel Nyondo[2], Alain Amstutz[3,4☯]*, Baptiste du Reau de la Gaignonnière[1,5☯]

1 Seboche Mission Hospital, Seboche, Lesotho, 2 Charlotte Maxeke Johannesburg Academic, University of the Witwatersrand, Division of Cardiothoracic Surgery, Johannesburg, South Africa, 3 University of Basel, Basel, Switzerland, 4 CLEAR Methods Center, Division of Clinical Epidemiology, Department of Clinical Research, University Hospital Basel, Basel, Switzerland, 5 Hospital Center Lavaur, Lavaur, France

☯ These authors contributed equally to this work.
* alain.amstutz@unibas.ch

**Data Availability Statement:** The data relevant to this study are available from Zenodo at DOI: 10.

## Abstract

### Background

In 2019, 600'000 people in Africa died of heart failure and heart diseases will increase on the continent. It is crucial to understand the regional etiologies and risk factors for heart failure and underlying heart diseases. However, echocardiography data from rural Africa are scarce and from Lesotho non-existent. This study aims to examine the occurrence, characteristics and etiology of heart failure and heart diseases using echocardiography data from a referral hospital in rural Lesotho.

### Methods

We conducted a retrospective cohort study at Seboche Mission Hospital, the only referral hospital in Butha-Buthe district (Lesotho) with an echocardiography department. We included data from all individuals referred to the department between January 2020 and May 2021. From non-hospitalized patients echocardiographic diagnosis, sex and age were available, from hospitalized patients additional sociodemographic and clinical data could be extracted.

### Results

In the study period, a total of 352 echocardiograms were conducted; 213 had abnormal findings (among them 3 children). The majority of adult participants (130/210; 64%) were female and most frequent heart diseases were hypertensive (62/210, 30%), valvular (39/210, 19%) and chronic pulmonary (37/210, 18%). Heart failure represented 11% of hospitalizations in the same period. Among the 126 hospitalized heart failure patients, the most common etiology was chronic pulmonary heart disease (32/126; 25%). Former mine workers and people with a history of tuberculosis were more likely to have a chronic pulmonary heart disease.

5281/zenodo.7333586 (https://doi.org/10.5281/
zenodo.7333586).

**Funding:** Alain Amstutz received his salary through
a grant from the MD–PhD programme of the Swiss
National Science Foundation (grant number
323530_177576). No further funding was involved.
The funders had no role in study design, data
collection and analysis, decision to publish, or
preparation of the manuscript.

**Competing interests:** The authors have declared
that no competing interests exist.

## Conclusions

The leading cause of heart disease in this setting is hypertension. However, in contrast to
other African epidemiological studies, chronic pulmonary heart disease is unexpectedly
common. There is an urgent need to improve awareness and knowledge about lung dis-
eases, make diagnostic and therapeutic options available and increase prevention.

## Introduction

During the last two decades, cardiovascular diseases have rapidly emerged as a major cause of
disease and death in Africa [1]. According to the World Health Organization, cardiovascular
diseases are among the top five causes of death in Africa. Almost 1.1 million people died
because of cardiovascular diseases in Africa in 2019 [2]. Ischemic heart disease, stroke, and
hypertensive heart diseases are identified as the three most common causes of cardiovascular
death in Sub-Saharan Africa (SSA) [2].

The heart failure syndrome has been recognized as a significant contributor to cardiovascu-
lar disease burden in SSA for many decades. The increasing burden of heart failure in the
region is driven by increasing urbanization, changes in lifestyle habits (decreased physical
activity, increased alcohol and nicotine use), and ageing of the population, and thus, in a surge
of hypertension, type 2 diabetes mellitus, dyslipidemia and obesity [3]. Mortality of patients
with heart failure has been reported to be highest in Africa, compared to other low-and mid-
dle-income regions; in Africa alone the mortality of patients with heart failure was estimated at
34% [4]. The most commonly reported etiology of heart failure in low-and middle-income
regions is ischemic heart disease, except for South America and Africa [5]. A recent meta-anal-
ysis from Africa including 10,098 patients from 22 studies found not ischemic/coronary heart
disease but hypertensive heart disease to be the commonest cause of heart failure at 39.2%, fol-
lowed by cardiomyopathies (21.4%), rheumatic heart disease (14.1%), and ischemic/coronary
heart disease only at 7.2% [6]. However, data from low- and middle-income countries are
scarce due to the non-availability of routine echocardiography, especially in rural areas.

The WHO estimates that cardiovascular diseases represent 14% of premature deaths due to
non-communicable diseases in Lesotho [7]. Detailed data on heart diseases and heart failure in
Lesotho, however, are non-existent. In this study, we describe the occurrence, characteristics,
and etiology of heart diseases and heart failure, diagnosed using echocardiography, in patients
from a rural referral hospital in Lesotho.

## Methods

### Study design

We conducted a retrospective cohort study at Seboche Mission Hospital. The secondary-level
referral hospital is situated in Butha-Buthe district, Northern Lesotho, on 1800 meters above
sea level. It is a public missionary hospital and serves a rural local population (Basotho), mostly
subsistence farmers and migration workers. Since 2020, the hospital has an established echo-
cardiography department that performs about 20–30 echocardiographic assessments (includ-
ing electrocardiograms) per month and represents the only echocardiographic referral point
for the entire district.

We included retrospective data of all individuals–children <18 years old and adults–
referred to the echocardiography department between January 2020 and May 2021 due to

signs and symptoms of heart failure. From patients that were hospitalized detailed sociodemographic and clinical data was available, whereas for the others only sex and age.

## Data collection

We extracted data from in-patient routine medical records, the echocardiography department patient register and the echocardiography reports. The participants' demographics, clinical information, echocardiography findings and laboratory values were entered into an electronic data capture spreadsheet by the physician and echocardiographer. Missing demographic information was collected through telephonic consultation with the patient. The data was double-checked by an external collaborator.

## Definitions

We provide detailed information about the echocardiography machine, techniques and measurements used in S1 Text. We adhered to the following clinical and echocardiographic definitions:

**Heart failure.**   Clinical syndrome of effort intolerance related to an abnormality of cardiac function, characterized by typical symptoms of shortness of breath, fatigue and leg swelling and accompanied by clinical signs of congestion, such as peripheral oedema, elevated jugular venous pressure and pulmonary crepitations [8].

**Functional heart failure.**   Heart failure signs without cardiac abnormality, caused by another reason (e.g. severe anemia, hyperthyroidism).

**Right heart failure.**   Tricuspid Annular Plan Systolic Excursion < 17 mm or RV systolic excursion velocity by tissue doppler imaging < 9.5 cm/s or fractional area change < 32% [9].

**Hypertensive heart disease.**   Left ventricular (LV) hypertrophy or concentric remodeling (normal LV mass with a relative wall thickness > 0.42) with or without global systolic or diastolic left ventricular dysfunction in a patient with arterial hypertension (systolic blood pressure >140mmHg or diastolic blood pressure >90mmHg or presence of antihypertensive therapy), with neither valve disease nor segmental wall motion abnormalities [8].

**Hypertrophic cardiomyopathy.**   A maximal end-diastolic wall thickness of $\geq$15 mm anywhere in the LV, in the absence of another cause of hypertrophy [10].

**Hypertrophy of the LV in children.**   LV mass/height$^{2.7}$ above the 95[th] percentile; severe LV hypertrophy: LV mass of at least 30% above the 95[th] percentile [11].

**Dilated cardiomyopathy.**   LV or biventricular dilation and impaired contraction, not explained by abnormal loading conditions (e.g. hypertension and valvular heart disease) or coronary artery disease [12].

**Valvular heart disease.**   Abnormal size and/or function of the heart and a primary abnormality of a valve (i.e., presence of valve regurgitation or stenosis and thickening of cusps, leaflets, or leaflet tips, vegetations or ruptured chordae tendineae). As a subgroup of the valvular heart diseases, rheumatic heart disease was defined according to the 2012 world heart federation criteria [13].

**Coronary heart disease.**   Typical angina pectoris and ventricular dysfunction with segmental hypo- or akinesia which could be attributed to a specific coronary artery with or without typical ECG findings [8].

**Pulmonary heart disease.**   Right heart failure in presence of pulmonary hypertension and normal left atrium pressure, sub-divided into either chronic or acute pulmonary heart disease [8].

**Pericardial heart disease.**   Pericardial effusion as the primary reason for the heart failure. Tuberculosis (TB) pericarditis if clinically suspected or microbiologically confirmed TB [8].

**Rhythmic heart disease.**    Congestive heart failure caused by rhythmic abnormalities (i.e., fast atrial fibrillation, complete auriculo-ventricular block) [8].

**Congenital heart disease.**    Definition according to the American Society of Echocardiography Pediatric and Congenital Heart Disease Council [14].

**Peri-/Post-partum cardiomyopathy.**    Cardiomyopathy with a reduced left ventricular ejection fraction (LVEF) of <45%, presenting towards the end of the pregnancy or in the months after delivery in a woman without previously known structural heart disease [15].

## Statistical analysis

We used absolute and relative frequencies to describe categorical data and medians and interquartile ranges for continuous variables. Inferential statistical testing to investigate factors predicting the most common etiology of heart failure was conducted using univariate and multivariate logistic regression models. We applied two-sided p-values with alpha 0.05 level of significance and presented the results as odds ratios with 95% confidence intervals. Descriptive statistics were conducted using Microsoft Excel and inferential statistics using Stata (version 14, Stata Corporation, Austin/Texas, USA).

## Ethics statement

The National Health Research and Ethics Committee of the Ministry of Health of Lesotho reviewed the study protocol and concluded that no written informed consent is needed since this is a retrospective study only involving the collection of existing data, registers and documents (ID146-2021; August 06, 2021).

## Results

Between January 2020 and May 2021, a total of 352 echocardiograms were conducted at the echocardiography department of Seboche Mission Hospital. Of the 352 echocardiograms, 11 (3%) were follow-up assessments. Among the 341 included echocardiograms (335 adults and 6 children), 128 (38%) indicated normal or non-significant findings, e.g. a minimal mitral valve regurgitation without any other sign of a heart disease (Table 1).

### Heart diseases: Occurrence, characteristics and etiology

Among the 213 individuals with abnormal findings, 3 were children (all with a congenital heart disease), and the remaining 210 participants were adults, of whom the majority (130/210; 64%) were female and were a median age of 62 (interquartile range [IQR] 52–75) years old (Table 1). The most frequent heart diseases among adults were hypertensive (62/210, 30%), valvular (39/210, 19%), chronic pulmonary (37/210, 18%), dilated cardiomyopathy (26/210, 12%), and coronary heart disease (17/210, 8%), followed by the remaining etiologies below 5% prevalence (Table 1 and Fig 1).

### Heart failure: Occurrence, characteristics and etiology

Among the 210 adult individuals with abnormal findings, 118 were hospitalized due to heart failure. In the same time period, in addition, eight adult patients with functional heart failure, i.e. caused by another reason than cardiac abnormality (e.g. severe anemia), were hospitalized. Overall, at Seboche Mission hospital between January 2020 and Mai 2021, hospitalization due to heart failure represented 11% (126/1164) of all hospitalized patients. The majority of patients hospitalized due to heart failure were female (56%; 70/126) and had a median age of 66 years old (IQR 54–76). 61% (77/126) had arterial hypertension, 65% (82/126) took cardiac

**Table 1. Characteristics of all patients referred for echocardiography.**

| Adults (n = 335) | | |
|---|---|---|
| | Total (N = 335) | Abnormal echocardiogram (n = 210) | Normal echocardiogram (n = 125) |
| **Sex** | | | |
| female | 221 (66%) | 130 (62%) | 91 (73%) |
| male | 114 (34%) | 80 (38%) | 34 (27%) |
| **Age, median (IQR)** | 58 (46–72) | 62 (52–75) | 50 (33–65) |
| | Only Adults with Abnormal Echocardiogram, by sex (n = 210) | | |
| | Total (N = 210) | Women (n = 130) | Men (n = 80) |
| **Hypertensive HD** | 62 (30%) | 48 (37%) | 14 (18%) |
| **Valvular HD** | 39 (19%) | 28 (22%) | 11 (14%) |
| **Chronic pulmonary HD** | 37 (18%) | 7 (5%) | 30 (38%) |
| **Dilated CM** | 26 (12%) | 17 (13%) | 9 (11%) |
| **Coronary HD** | 17 (8%) | 11 (8%) | 6 (8%) |
| **Acute pulmonary HD** | 9 (4%) | 6 (5%) | 3 (4%) |
| **Pericardial HD** | 9 (4%) | 6 (5%) | 3 (4%) |
| **Rhythmic HD** | 8 (4%) | 4 (3%) | 4 (5%) |
| **Hypertrophic CM** | 2 (1%) | 2 (2%) | 0 (0%) |
| **Unclassified CM** | 1 (0%) | 1 (1%) | 0 (0%) |
| | Children (n = 6) | | |
| | Total (N = 6) | Abnormal echocardiogram (n = 3) [1] | Normal echocardiogram (n = 3) |
| **Sex** | | | |
| female | 3 (50%) | 2 (67%) | 1 (33%) |
| male | 3 (50%) | 1 (33%) | 2 (67%) |
| **Age, median (IQR)** | 1 (1–2) | 2 (1–12) | 1 (1–1) |

Abbreviations: IQR (interquartile range), CM (cardiomyopathy), HD (heart disease)

[1] Congenital heart diseases: 1 girl with an aortic stenosis (bicuspid aortic valve) and 1 girl and 1 boy with a patent ductus arteriosus

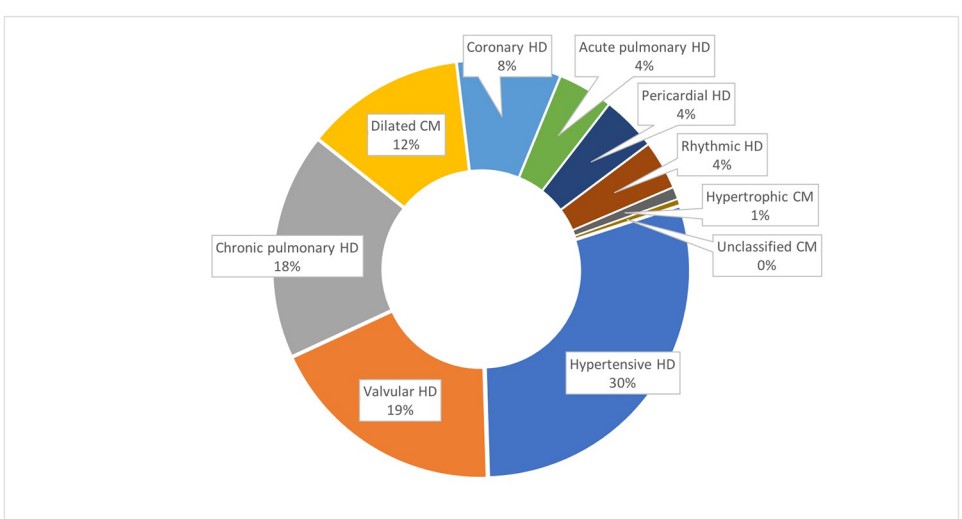

**Fig 1. Etiology of heart diseases among all adult patients with an abnormal echocardiogram (n = 210).**
Abbreviations: CM (cardiomyopathy), HD (heart disease).

**Table 2. Characteristics of patients hospitalized due to heart failure, by sex.**

| | Total (n = 126) | Male (n = 56; 44%) | Female (n = 70; 56%) |
|---|---|---|---|
| **Characteristics and cardiovascular risk factors** | | | |
| Age in years, median (IQR) | 66 (54–76) | 68 (56–76) | 66 (51–75) |
| Hypertensive | 77 (61%) | 20 (36%) | 57 (81%) |
| Diabetic | 25 (20%) | 6 (11%) | 19 (27%) |
| Living with HIV | 17 (13%) | 9 (16%) | 8 (11%) |
| History of tuberculosis | 19 (15%) | 17 (30%) | 2 (3%) |
| Takes cardiac medication | 82 (65%) | 25 (45%) | 57 (81%) |
| Nicotine smoking | 18 (14%) | 15 (27%) | 3 (4%) |
| Former mine worker | 29 (23%) | 28 (50%) | 1 (1%) |
| **Symptoms and clinical examination** | | | |
| Oedema | 118 (94%) | 52 (93%) | 66 (94%) |
| Dyspnea | 115 (91%) | 53 (95%) | 62 (89%) |
| Chest pain | 27 (21%) | 9 (16%) | 18 (26%) |
| Irregular heart beat | 23 (18%) | 5 (9%) | 18 (26%) |
| Systolic blood pressure, mmHg, median (IQR) | 132 (113–149) | 126 (107–144) | 138 (118–151) |
| Diastolic blood pressure, mmHg, median (IQR) | 79 (68–92) | 78 (68–92) | 80 (70–96) |
| SpO2, %, median (IQR) | 91 (80–95) | 88 (76–92) | 93 (87–96) |
| Haemoglobin, g/dL, median (IQR) | 13 (11–15) | 14 (11–16) | 12 (11–14) |

Abbreviations: IQR (interquartile range), CM (cardiomyopathy), heart disease (heart disease), heart failure (heart failure)

medication at presentation, 20% (25/126) were diabetic, 13% (17/126) lived with HIV and 15% (19/126) had a history of tuberculosis. A total of 14% (18/126) were smokers and 23% (29/126) former mine workers. During clinical examination, most presented with oedema (94%; 118/126) and dyspnea (91%; 115/126) (Table 2).

The detailed echocardiographic data of the patients hospitalized due to heart failure are listed in S1 Table. Only 48% (60/126) had a normal sized LV. Left ventricular systolic function was preserved in 60% (75/126) of participants, but severely impaired in 22% (28/126). 29% (36/126) of patients had a ventricular hypertrophy and 24% (30/126) presented with left ventricular dilatation. Moderate to severe valve regurgitation of the aortic valve accounted for 5% (6/126) of patients and of the mitral valve for 13% (16/126) of patients. 2% (2/126) of patients had a moderate to severe aortic valve stenosis and 3% (4/126) of patients a moderate to severe mitral valve stenosis. Severe tricuspid insufficiency was noted in 15% (19/126) of patients. 31% (39/126) of patients had some pericardial effusion, but in only 5% (6/126) of patients this effusion was the cause of the heart failure (S1 Table).

The most frequent etiology for heart failures were chronic pulmonary heart disease (32/126, 25%), followed by dilated cardiomyopathy (21/126, 17%) and valvular heart disease (20/126, 16%) (Fig 2). Among the valvular heart diseases, most were rheumatic origin (Table 3). Coronary heart diseases accounted for 11% (14/126) and hypertensive heart diseases represented 9% (11/126) of all reasons for hospitalization due to heart failure (Fig 2 and Table 3). 4 patients died during the hospitalization: A 70 years old male with chronic pulmonary heart disease, a 18 years old male with a fulminant acute pulmonary heart disease, a 56 years old female with dilated cardiomyopathy and a 51 years old female with a valvular (rheumatic) heart disease.

## Heart failure: Risk factors for the most common heart failure etiology

Chronic pulmonary heart disease was the most common etiology for heart failure among the hospitalized study population. Univariate logistic regression revealed that several patient

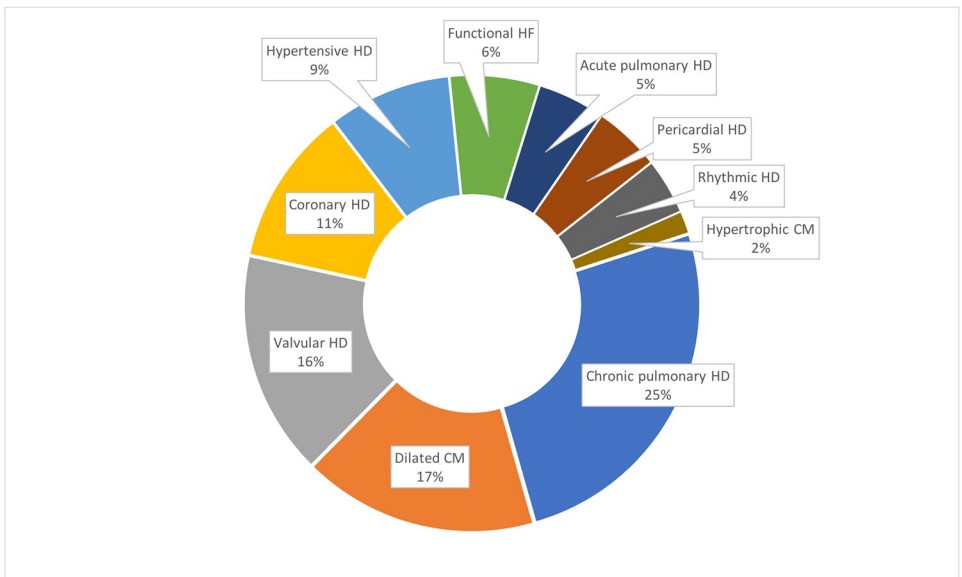

**Fig 2. Etiology of heart failure (n = 126).** Abbreviations: CM (cardiomyopathy), HD (heart disease), HF (heart failure).

characteristics were associated with chronic pulmonary heart disease with strongest associations for male, arterial hypertension, history of TB and former mine worker (S2 Table). When the multivariate logistic regression model was fitted with these variables, only a history of TB (adjusted odds ratio 6.25 [95% confidence interval 1.24–31.481]; p-value 0.026) and former mine worker (adjusted odds ratio 24.65 [95% confidence interval 5.06–120.101]; p-value <0.001) remained risk factors for chronic pulmonary heart disease.

**Table 3. Etiology of heart failure.**

| Etiology | Total (N = 126) | Male (n = 56) | Female (n = 70) |
|---|---|---|---|
| **Chronic pulmonary HD** | **32 (25%)** | **27 (48%)** | **5 (7%)** |
| **Dilated CM** | **21 (17%)** | **5 (9%)** | **16 (23%)** |
| Post-partum CM | 3 (2%) | 0 (0%) | 3 (4%) |
| **Valvular HD** | **20 (16%)** | **5 (9%)** | **15 (21%)** |
| Rheumatic HD | 12 (10%) | 4 (7%) | 8 (11%) |
| **Coronary HD** | **14 (11%)** | **5 (9%)** | **9 (13%)** |
| **Hypertensive HD** | **11 (9%)** | **5 (9%)** | **6 (9%)** |
| **Functional HF** | **8 (6%)** | **4 (7%)** | **4 (6%)** |
| **Acute pulmonary HD** | **6 (5%)** | **2 (4%)** | **4 (6%)** |
| **Pericardial HD** | **6 (5%)** | **2 (4%)** | **4 (6%)** |
| Tuberculosis pericarditis | 4 (3%) | 2 (4%) | 2 (3%) |
| Non-tuberculosis pericarditis | 2 (2%) | 0 (0%) | 2 (3%) |
| **Rhythmic HD** | **5 (4%)** | **1 (2%)** | **4 (6%)** |
| **Hypertrophic CM** | **2 (2%)** | **0 (0%)** | **2 (3%)** |
| **Unclassified CM** | **1 (1%)** | **0 (0%)** | **1 (1%)** |

Abbreviations: CM (cardiomyopathy), heart disease (heart disease), heart failure (heart failure)

## Discussion

This retrospective cohort study included all echocardiograms (n = 352) performed during a 17-month period in 2020/21 at a rural hospital in Northern Lesotho. Among the 210 adult participants with a heart disease, hypertensive heart disease (30%), valvular heart disease (19%), chronic pulmonary heart disease (18%), dilated CM (12%) and coronary heart disease (8%) were the most common etiologies. Heart failure represented a substantial burden of hospitalizations with 11% of all hospitalized patients in the same period. Among the 126 hospitalized participants with heart failure, only 48% had a normal sized LV and only 60% had a preserved left ventricular systolic function. The most common etiologies were chronic pulmonary heart disease (25%), dilated CM (17%), valvular heart disease (16%, most rheumatic), coronary heart disease (11%) and hypertensive heart disease (9%). Former mine workers and people with a history of TB were more likely to have a chronic pulmonary heart disease.

In 2019, about 600'000 people in Africa died of heart failure due to a heart disease [2]. It is estimated that the burden of heart diseases will increase in the coming decade and that the underdiagnosis rate is high [3]. In order to tackle this health problem, it is of paramount importance to understand the etiologies and risk factors for heart diseases and heart failure in the region.

In line with a recent meta-analysis of 22 African studies, hypertensive heart disease is the most common heart disease [6]. Moreover, valvular (mostly due to rheumatic origin) and cardiomyopathies are common etiologies, whereas ischemic or coronary heart diseases are–different to the rest of the world–still rather an infrequent reason at 7% of all heart failure cases, similar to our findings [6]. What is striking in our study, is the high number of chronic pulmonary heart diseases: It was the third most frequent heart disease (18%) among all assessed patients (hospitalized and non-hospitalized) and the most frequent heart disease (25%) among hospitalized patients with heart failure.

A comparable prospective echocardiography cohort study conducted at a rural referral hospital in Tanzania concluded that hypertensive heart disease (41%), followed by valvular heart disease (18%), coronary heart diseases (18%) and cardiomyopathies (15%) were the most common heart diseases [16]. Similar to the meta-analysis, they observed only 5% of pulmonary heart diseases. In contrast, a recent South African study of 119 patients with heart failure documented 12% pulmonary heart diseases, indicating, that there might be a regional difference in etiologies [17]. Chronic pulmonary heart disease or cor pulmonale is a right heart failure caused by long-term high blood pressure in the lung arteries and right ventricle, whereas the left heart works normal [18]. Most commonly it results from chronic lung diseases such as chronic obstructive pulmonary disease (COPD) [18]. The mining sector in Lesotho and South Africa is one of the main industries for Basotho, especially the men. It is well documented that Basotho mine workers have a high risk of lung diseases such as TB, COPD and silicosis [19, 20]. While working conditions in the official mines have improved, illegal mining among Basotho has increased, as many South African mines have closed down or become less accessible for foreign workers [21]. Another reason for increased chronic lung diseases and thus resulting in pulmonary heart diseases, might be the exposure to indoor air pollution from biomass cooking fuels, a common practice in rural Lesotho. A meta-analysis concludes strong association of this cooking practice with COPD with highest risk in the African region [22]. HIV infection itself is a cause of pulmonary hypertension, was a risk factor in the univariate logistic regression analysis, is prevalent in the setting and thus, may have been contributing to the high proportion of this kind of heart disease [23].

Treatment options for patients with chronic pulmonary heart diseases are limited and unspecific: Improvement of the underlying cause (e.g. inhaled pulmonary vasodilator therapy and oxygen), relief of symptoms (e.g. diuretics), and prevention of complications (e.g.

anticoagulation). While some of the study participants received diuretics, none were taking any medication to improve the lung function nor were anticoagulated. Awareness and knowledge about lung diseases and diagnostic equipment are scarce and anticoagulation is routinely not done due to expensive monitoring and medication.

Our study has several limitations. First, it is a retrospective cohort study based on available clinical data with the risk of sampling bias. That's the reason why we have incomplete data for non-hospitalized patients and had to focus our risk factor analysis on hospitalized patients, i.e. with acute heart failure. Nevertheless, the echocardiographic diagnosis of the heart diseases was systematically available for all study patients. Moreover, the heart failure syndrome contributes most to the mortality of heart diseases, it is important to focus on this population, as done in similar studies [6]. Second, in our setting coronary angiography, myocardial scintigraphy, stress-echocardiography and myocardial biopsies are not available. Thus, the differentiation of cardiomyopathies was not possible, and diagnosis of coronary heart disease was based on medical history, clinical examination, electrocardiogram and echocardiogram only. Third, we had no data on lung function, indoor air pollution or other cardiovascular risk factors. More research is warranted in this area.

To our knowledge, this is the first echocardiographic study from Lesotho systematically assessing the etiology of heart diseases and one of few in southern Africa. Another strength is that the study was able to determine the burden of the heart failure syndrome at a typical referral hospital in the region and evaluate the most frequent causes and its risk factors. The findings offer important insights for the region.

In conclusion, this study established the causes of heart diseases and heart failure, which is important for prevention and subsequent clinical management. The leading cause of heart disease in this setting is hypertension and it is thus crucial to improve prevention, screening and treatment of hypertension. However, in contrast to (northern) African epidemiological studies, pulmonary heart disease is unexpectedly common. Former mine workers and people with a history of TB were more likely to have a chronic pulmonary heart disease. There is an urgent need to improve awareness and knowledge about lung diseases among the community as well as clinicians, make diagnostic and therapeutic options available and increase prevention measures to reduce TB and air pollution exposure in the mines and homes of people in southern Africa.

## Supporting information

**S1 Text. Echocardiography details.**
(DOCX)

**S1 Table. Echocardiography findings of patients hospitalized due to heart failure.**
(DOCX)

**S2 Table. Association between characteristics and most common etiology for heart failure.**
(DOCX)

## Acknowledgments

We would like to recognize the hard work of the staff at Seboche Mission Hospital and most importantly, we gratefully acknowledge the study participants.

## Author Contributions

**Conceptualization:** Nzola John Ndongala, Callixtina Maepa, Emmanuel Nyondo, Alain Amstutz, Baptiste du Reau de la Gaignonnière.

**Data curation:** Nzola John Ndongala, Callixtina Maepa, Alain Amstutz, Baptiste du Reau de la Gaignonnière.

**Formal analysis:** Nzola John Ndongala, Alain Amstutz, Baptiste du Reau de la Gaignonnière.

**Funding acquisition:** Alain Amstutz.

**Investigation:** Nzola John Ndongala, Emmanuel Nyondo, Baptiste du Reau de la Gaignonnière.

**Methodology:** Nzola John Ndongala, Alain Amstutz, Baptiste du Reau de la Gaignonnière.

**Project administration:** Callixtina Maepa, Emmanuel Nyondo, Baptiste du Reau de la Gaignonnière.

**Resources:** Callixtina Maepa, Emmanuel Nyondo.

**Software:** Baptiste du Reau de la Gaignonnière.

**Supervision:** Emmanuel Nyondo, Alain Amstutz, Baptiste du Reau de la Gaignonnière.

**Validation:** Alain Amstutz, Baptiste du Reau de la Gaignonnière.

**Visualization:** Alain Amstutz.

**Writing – original draft:** Nzola John Ndongala, Alain Amstutz.

**Writing – review & editing:** Callixtina Maepa, Emmanuel Nyondo, Alain Amstutz, Baptiste du Reau de la Gaignonnière.

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
