## [Decision Letter · Decision Letter 0]

12 Jul 2022

PONE-D-21-39404Etiology, characteristics and occurrence of heart diseases in rural Lesotho (ECHO-Lesotho): A retrospective echocardiography cohort studyPLOS ONE

Dear Dr. Amstutz,

Thank you for submitting your manuscript to PLOS ONE. After careful consideration, we feel that it has merit but does not fully meet PLOS ONE’s publication criteria as it currently stands. Therefore, we invite you to submit a revised version of the manuscript that addresses the points raised during the review process.

We look forward to receiving your revised manuscript.

Kind regards,

Lucio Careddu, Ph.D.

Academic Editor

PLOS ONE

Journal Requirements:

Reviewers' comments:

Reviewer's Responses to Questions

**Comments to the Author**

1. Is the manuscript technically sound, and do the data support the conclusions?

Reviewer #1: Yes

Reviewer #2: Yes

2. Has the statistical analysis been performed appropriately and rigorously? 

Reviewer #1: Yes

Reviewer #2: Yes

3. Have the authors made all data underlying the findings in their manuscript fully available?

Reviewer #1: Yes

Reviewer #2: Yes

4. Is the manuscript presented in an intelligible fashion and written in standard English?

Reviewer #1: Yes

Reviewer #2: Yes

5. Review Comments to the Author

Reviewer #1: The authors conducted a retrospective cohort study for adult access to cardiology department in order to identify the leading cause of heart disease in this population

Interestingly, heart failure represented a substantial burden of hospitalizations and the most common etiologies were chronic pulmonary heart disease with former mine workers and people with a history of TB being more likely to have a chronic pulmonary heart disease

In order to tackle this health problem, the authors underline how it is paramount importance to understand the etiologies and risk factors for heart diseases and heart failure in the region.

Moreover, the high number of chronic pulmonary heart diseases is striking and it is relevant how more research is warranted in this area.

The definitions of hypertrophic cardiomyopathy and dilated cardiomiopathy aren’t correct.

A clinical diagnosis of HCM in adult patients can therefore be established by imaging with 2D echocardiography or cardiovascular magnetic resonance (CMR) showing a maximal end-diastolic wall thickness of ≥15 mm anywhere in the left ventricle, in the absence of another cause of hypertrophy in adults (https://doi.org/10.1161/CIR.0000000000000937)

Dilated Cardiomyopathy is defined as presence of left ventricular dilatation and left ventricular systolic dysfunction in the absence of abnormal loading conditions (hypertension and valve disease) or coronary artery disease sufficient to cause global systolic impairment (J Am Soc Echocardiogr 2015;28:1-39)

Reviewer #2: I read with pleasure the manuscript “Etiology, characteristics and occurrence of heart diseases in rural Lesotho (ECHO-Lesotho): a retrospective echocardiography cohort study”, which aims to examine the occurrence, etiology and characteristics of heart diseases in Lesotho analyzing the echocardiographic data of a referral hospital.

The study is very interesting because currently in the literature there are no data on heart diseases in Lesotho and, taking into account that cardiovascular diseases are becoming a major cause of death in Sub-Saharan Africa, it is important that they are investigated.

However, I have to point out some issues that, in my opinion, need to be changed in order to accept and publish the scientific article:

- The study has a major limitation regarding the small number of patients. The data collected from January 2020 to May 2021 are not sufficient to describe the situation of heart diseases in an African State. Data collection should be extended retrospectively as much as possible or, if the echocardiographic laboratory was recently established, it could be useful to combine the data with those of another hospital in Lesotho.

- All the data collected with the echocardiogram were carefully described in the section “Echocardiography”, as well as the techniques used, but none of these data were reported in the manuscript. I don’t think it is necessary to describe these data so accurately for the purposes of the study that is being carried out and especially if they should not be included in the scientific article.

- In the section “Definitions”, it would be interesting to insert the guideline or bibliographic reference for each definition.

- In the table 1 there are incorrect numbers: the sum of females (n. 134) and males (n.84) with abnormal echocardiogram is 218 instead of 210, the sum of females (n. 87) and males (n. 30) with normal echocardiogram is 117 instead of 125.

- In addition, in the table 1 it is also necessary to specify the different type of congenital heart disease diagnoses in the children with an abnormal echocardiogram.

- In the section “Heart disease: Occurrence, characteristics and etiology”, is the sentence “of whom the majority (130/210; 64%) were female with a median age of 58 (interquartile range [IQR] 46-72) years old” correct? The data does not match with the data of table 1.

- In the section “Heart failure: Occurrence, characteristics and etiology”, is the sentence “the majority of patients hospitalized due to heart failure were female (56%; 70/126) and had a median age of 66 years old (IQR 54-76)” correct? The data does not match with the data of table 2.

- In the table 2 it would be better to divide the section of cardiovascular risk, the section of symptoms and the section of vital signs for greater clarity.

- In the supplement table 1, the sum of percentages in section LV size, LV systolic function, diastolic relaxation impairment of LV is 101% instead 100%.

- In addition, in the supplement table 1, regarding the LV systolic function and in particular the mild systolic heart failure, replace “LVEF 41-53%” with “LVEF 41-54%”.

- In the line 263 “table 3” should be changed with “supplement table 1”.

- During hospitalization for heart failure, which was the mean length of stay? Have there been any cardiovascular death and which were the causes? Could you add this additional data in the article?

6. PLOS authors have the option to publish the peer review history of their article (what does this mean?). If published, this will include your full peer review and any attached files.

Reviewer #1: No

Reviewer #2: No

---

## [Author Response · Author response to Decision Letter 0]

1 Aug 2022

Find the well-formatted response letter uploaded separately.

Response to Reviewers

Etiology, characteristics and occurrence of heart diseases in rural Lesotho (ECHO-Lesotho): A retrospective echocardiography cohort study

Journal Requirements:

Response: We updated our manuscript to meet PLOS ONE’s style requirements

Response: We updated our funding information and provided the grant number in the in all relevant sections

Response: We are already preparing the anonymized dataset for upload onto Zenodo and will be ready with DOI in time.

Response: We updated our manuscript to meet PLOS ONE’s style requirements

Reviewer #1: 

1. The definitions of hypertrophic cardiomyopathy and dilated cardiomiopathy aren’t correct. A clinical diagnosis of HCM in adult patients can therefore be established by imaging with 2D echocardiography or cardiovascular magnetic resonance (CMR) showing a maximal end-diastolic wall thickness of ≥15 mm anywhere in the left ventricle, in the absence of another cause of hypertrophy in adults (https://doi.org/10.1161/CIR.0000000000000937). Dilated Cardiomyopathy is defined as presence of left ventricular dilatation and left ventricular systolic dysfunction in the absence of abnormal loading conditions (hypertension and valve disease) or coronary artery disease sufficient to cause global systolic impairment (J Am Soc Echocardiogr 2015;28:1-39)

Response: Thank you very much for spotting this important error from our side. We realized that there was a misunderstanding between the writing and the clinical team and the definitions for HCM and DCM were not correctly reflected. We adapted the definitions, added the corresponding guideline references and these were the definitions used by the clinical team during examination. We adapted accordingly in section “Definitions”: 

Dilated cardiomyopathy. “LV or biventricular dilation and impaired contraction, not explained by abnormal loading conditions (e.g. hypertension and valvular heart disease) or coronary artery disease”

Hypertrophic cardiomyopathy. “A maximal end-diastolic wall thickness of ≥15 mm anywhere in the LV, in the absence of another cause of hypertrophy”

Reviewer #2: 

1. I read with pleasure the manuscript “Etiology, characteristics and occurrence of heart diseases in rural Lesotho (ECHO-Lesotho): a retrospective echocardiography cohort study”, which aims to examine the occurrence, etiology and characteristics of heart diseases in Lesotho analyzing the echocardiographic data of a referral hospital. The study is very interesting because currently in the literature there are no data on heart diseases in Lesotho and, taking into account that cardiovascular diseases are becoming a major cause of death in Sub-Saharan Africa, it is important that they are investigated.

Response: Thank you very much for your positive feedback.

2. However, I have to point out some issues that, in my opinion, need to be changed in order to accept and publish the scientific article: The study has a major limitation regarding the small number of patients. The data collected from January 2020 to May 2021 are not sufficient to describe the situation of heart diseases in an African State. Data collection should be extended retrospectively as much as possible, or, if the echocardiographic laboratory was recently established, it could be useful to combine the data with those of another hospital in Lesotho.

Response: We agree with the reviewer that the data is limited. The echocardiography department was established in January 2020 (mentioned in the manuscript). The only other health facility in Lesotho that has an echocardiography department is in the capital, in Maseru. During the implementation of the echocardiography department, we were in contact with that hospital, however, they did not agree to share their data. We are planning to conduct an update of our data, but since there was a change of leadership in mid-2021 it became more difficult to collaborate. Despite these shortcomings, we believe that it is important to get this data out into the public – to at least have some initial data from Lesotho – and to stimulate more research about this topic in the country.

3. All the data collected with the echocardiogram were carefully described in the section “Echocardiography”, as well as the techniques used, but none of these data were reported in the manuscript. I don’t think it is necessary to describe these data so accurately for the purposes of the study that is being carried out and especially if they should not be included in the scientific article.

Response: We think that these details are important for two reasons. First, these details help to understand the echocardiography definitions we used and the results presented (Supporting Information S1 Table provides the details). Second, we think these details are crucial in case someone wants to reproduce our study findings. However, we understand the reviewers concern and think this paragraph is too prominent in the main manuscript. Thus, we moved this entire paragraph into the Supporting Information (S1 Text) and added a sentence in section “Definitions”:

“We provide detailed information about the echocardiography machine, techniques and measurements used in S1 Text.”

4. In the section “Definitions”, it would be interesting to insert the guideline or bibliographic reference for each definition.

Response: We added the respective guideline/reference for each definition.

5. In the table 1 there are incorrect numbers: the sum of females (n. 134) and males (n.84) with abnormal echocardiogram is 218 instead of 210, the sum of females (n. 87) and males (n. 30) with normal echocardiogram is 117 instead of 125.

Response: Thank you for pointing out this important typo. We corrected the numbers. 

6. In addition, in the table 1 it is also necessary to specify the different type of congenital heart disease diagnoses in the children with an abnormal echocardiogram.

Response: We added the details of these 3 congenital heart diseases in the footnote of Table 1:

“[1] Congenital heart diseases: 1 girl with an aortic stenosis (bicuspid aortic valve) and 1 girl and 1 boy with a patent ductus arteriosus”

7. In the section “Heart disease: Occurrence, characteristics and etiology”, is the sentence “of whom the majority (130/210; 64%) were female with a median age of 58 (interquartile range [IQR] 46-72) years old” correct? The data does not match with the data of table 1.

Response: Thank you for spotting this typo. It was related to the above typo and the median age was wrongly taken from the overall population instead of only the population with abnormal echocardiogram. We corrected the numbers.

8. In the section “Heart failure: Occurrence, characteristics and etiology”, is the sentence “the majority of patients hospitalized due to heart failure were female (56%; 70/126) and had a median age of 66 years old (IQR 54-76)” correct? The data does not match with the data of table 2.

Response: This is correct and matches the data of table 2. There were a total of 126 hospitalized patients due to heart failure (Total in first column), they had a median age of 66 years with an IQR 54-76 (median age in first column) and 56% were female (n=70/126, third column).

9. In the table 2 it would be better to divide the section of cardiovascular risk, the section of symptoms and the section of vital signs for greater clarity.

Response: We thank the reviewer for this great suggestion and adapted accordingly. 

10. In the supplement table 1, the sum of percentages in section LV size, LV systolic function, diastolic relaxation impairment of LV is 101% instead 100%.

Response: This was due to rounding. We added one decimal digit everywhere to avoid misleading sums due to rounding.

11. In addition, in the supplement table 1, regarding the LV systolic function and in particular the mild systolic heart failure, replace “LVEF 41-53%” with “LVEF 41-54%”.

Response: Thank you for spotting this. The mistake was not adding a ‘greater than or equal sign’ in front of LVEF 54%. We corrected accordingly to “LVEF ≥ 54%”.

12. In the line 263 “table 3” should be changed with “supplement table 1”.

Response: We adapted accordingly.

13. During hospitalization for heart failure, which was the mean length of stay? Have there been any cardiovascular death and which were the causes? Could you add this additional data in the article?

Response: Unfortunately, our retrospective data collection tool did not include length of hospitalization. However, we do have the mortality data. We added a short paragraph just before section “Heart failure: risk factors for the most common heart failure etiology” to summarize this data:

“4 patients died during the hospitalization: A 70 years old male with chronic pulmonary heart disease, a 18 years old male with a fulminant acute pulmonary heart disease, a 56 years old female with dilated cardiomyopathy and a 51 years old female with a valvular (rheumatic) heart disease.”

---

## [Decision Letter · Decision Letter 1]

16 Nov 2022

Etiology, characteristics and occurrence of heart diseases in rural Lesotho (ECHO-Lesotho): A retrospective echocardiography cohort study

PONE-D-21-39404R1

Dear Dr. Amstutz,

We’re pleased to inform you that your manuscript has been judged scientifically suitable for publication and will be formally accepted for publication once it meets all outstanding technical requirements.

Kind regards,

Roberto Magalhães Saraiva, MD, PhD

Academic Editor

PLOS ONE

Additional Editor Comments (optional):

Reviewers' comments:

Reviewer's Responses to Questions

**Comments to the Author**

1. If the authors have adequately addressed your comments raised in a previous round of review and you feel that this manuscript is now acceptable for publication, you may indicate that here to bypass the “Comments to the Author” section, enter your conflict of interest statement in the “Confidential to Editor” section, and submit your "Accept" recommendation.

Reviewer #2: All comments have been addressed

2. Is the manuscript technically sound, and do the data support the conclusions?

Reviewer #2: Yes

3. Has the statistical analysis been performed appropriately and rigorously? 

Reviewer #2: Yes

4. Have the authors made all data underlying the findings in their manuscript fully available?

Reviewer #2: Yes

5. Is the manuscript presented in an intelligible fashion and written in standard English?

Reviewer #2: Yes

6. Review Comments to the Author

Reviewer #2: I read the reviewed version of the article “Etiology, characteristics and occurrence of heart diseases in rural Lesotho (ECHO-Lesotho): a retrospective echocardiography cohort study”, a retrospective cohort study which treats etiology, characteristics and occurrence of heart disease in Lesotho analyzing data from an echocardiography department. The comments raised in the previous review have been addressed as far as possible and, in my opinion, now this manuscript is acceptable for publication, aware of the fact that the small number of patients cannot fully represent the situation of heart disease in an African State. Maybe the article title should be changed to “a preview of a retrospective echocardiography cohort study” hinting at the intention to continue the study increasing the number of patients.

7. PLOS authors have the option to publish the peer review history of their article (what does this mean?). If published, this will include your full peer review and any attached files.

Reviewer #2: No

---

## [Editor Report · Acceptance letter]

7 Dec 2022

PONE-D-21-39404R1 

Etiology, characteristics and occurrence of heart diseases in rural Lesotho (ECHO-Lesotho): A retrospective echocardiography cohort study 

Dear Dr. Amstutz:

I'm pleased to inform you that your manuscript has been deemed suitable for publication in PLOS ONE. Congratulations! Your manuscript is now with our production department. 

Kind regards, 

on behalf of

Dr. Roberto Magalhães Saraiva 

Academic Editor

PLOS ONE